# Prevalence and Potential Determinants of COVID-19 Vaccine Hesitancy and Resistance in Qatar: Results from a Nationally Representative Survey of Qatari Nationals and Migrants between December 2020 and January 2021

**DOI:** 10.3390/vaccines9050471

**Published:** 2021-05-07

**Authors:** Salma M. Khaled, Catalina Petcu, Lina Bader, Iman Amro, Aisha Mohammed H. A. Al-Hamadi, Marwa Al Assi, Amal Awadalla Mohamed Ali, Kien Le Trung, Abdoulaye Diop, Tarek Bellaj, Mohamed H. Al-Thani, Peter W. Woodruff, Majid Alabdulla, Peter M. Haddad

**Affiliations:** 1Social and Economic Survey Research Institute, Qatar University, Doha P.O. Box 2713, Qatar; cpetcu@qu.edu.qa (C.P.); lina.bader@qu.edu.qa (L.B.); iman.amro@qu.edu.qa (I.A.); aisha.alhamadi@qu.edu.qa (A.M.H.A.A.-H.); marwa.alassi@qu.edu.qa (M.A.A.); amal.ali@qu.edu.qa (A.A.M.A.); kienle@qu.edu.qa (K.L.T.); adiop@qu.edu.qa (A.D.); 2Department of Public Health, College of Health Sciences, Qatar University, Doha P.O. Box 2713, Qatar; 3Department of Population Medicine, College of Medicine, Qatar University, Doha P.O. Box 2713, Qatar; 4College of Art and Sciences, Qatar University, Doha P.O. Box 2713, Qatar; tbellaj@qu.edu.qa; 5Department of Public Health, Ministry of Public Health, Al Khaleej Street, Rumaila, Doha P.O. Box 42, Qatar; malthani@moph.gov.qa; 6Department of Neuroscience, University of Sheffield, The University of Sheffield Western Bank, Sheffield S10 2TN, UK; p.w.woodruff@sheffield.ac.uk; 7Department of Psychiatry, Hamad Medical Corporation, Doha P.O Box 3050, Qatar; malabdulla3@hamad.qa (M.A.); phaddad@hamad.qa (P.M.H.); 8Clinical Science Department, College of Medicine, Qatar University, Doha P.O. Box 2713, Qatar

**Keywords:** COVID-19, vaccine willingness, hesitancy or refusal, Middle East and North Africa (MENA), Arab, migrant, Qatar

## Abstract

Global COVID-19 pandemic containment necessitates understanding the risk of hesitance or resistance to vaccine uptake in different populations. The Middle East and North Africa currently lack vital representative vaccine hesitancy data. We conducted the first representative national phone survey among the adult population of Qatar, between December 2020 and January 2021, to estimate the prevalence and identify potential determinants of vaccine willingness: acceptance (strongly agree), resistance (strongly disagree), and hesitance (somewhat agree, neutral, somewhat disagree). Bivariate and multinomial logistic regression models estimated associations between willingness groups and fifteen variables. In the total sample, 42.7% (95% CI: 39.5–46.1) were accepting, 45.2% (95% CI: 41.9–48.4) hesitant, and 12.1% (95% CI: 10.1–14.4) resistant. Vaccine resistant compared with hesistant and accepting groups reported no endorsement source will increase vaccine confidence (58.9% vs. 5.6% vs. 0.2%, respectively). Female gender, Arab ethnicity, migrant status/type, and vaccine side-effects concerns were associated with hesitancy and resistance. COVID-19 related bereavement, infection, and quarantine status were not significantly associated with any willingness group. Absence of or lack of concern about contracting the virus was solely associated with resistance. COVID-19 vaccine resistance, hesitance, and side-effects concerns are high in Qatar’s population compared with those globally. Urgent public health engagement should focus on women, Qataris (non-migrants), and those of Arab ethnicity.

## 1. Introduction

COVID-19, caused by the SARS-CoV-2 virus, is the most serious pandemic in living memory. Since the first case was reported in Wuhan in December 2019 [1], the virus has spread to 219 countries, infected over 100 million people, and claimed 2.4 million lives [2]. In an attempt to control the spread of infection countries have adopted quarantine and lockdown measures leading to huge social disruption. A global economic recession is inevitable [3]. ‘Herd’ or population immunity refers to the indirect protection from an infectious disease that occurs when the proportion of the population that is immune is sufficient to prevent sustained transmission. The immunity threshold for SARS-CoV-2 required for herd immunity is uncertain but several studies have suggested that it lies between 71–74% [4,5,6], though the recent appearance of more infectious variants may increase this figure [7]. Achieving herd immunity to COVID-19 through natural infections is not an option as it would entail an immense death toll, place an unacceptable strain on health services and require long-term social restrictions. The World Health Organization (WHO) supports reaching herd immunity through vaccination [8].

In December 2020, the Food and Drug Administration (FDA) issued the first emergency use authorization for a COVID-19 vaccine [9]. Since then vaccines have been approved by regulatory authorities around the world with further vaccines in development [10,11]. Vaccine development usually takes 5 to 18 years [12]. The development and approval of COVID-19 vaccines in under a year is an extraordinary scientific achievement [13,14]. Vaccines have the potential to save many lives and help bring the pandemic under control. However, they are not a panacea; the duration of protection conferred is unclear and further vaccinations may be necessary, analogous to the situation with influenza vaccination.

The benefits of COVID-19 vaccines can only be realized if a high proportion of the population accept the vaccine. The SAGE Working Group on Vaccine Hesitancy defined vaccine hesitancy as a “delay in acceptance or refusal of vaccines despite the availability of vaccination services” [15]. Vaccine hesitancy is not a new phenomenon, it was recognized with smallpox vaccination in the second half of the 19th century [16,17]. Studies from the USA and Europe show that although the majority of people are willing to accept COVID-19 vaccination, a significant proportion are ambivalent with another subgroup being firmly opposed to vaccination. A nationally representative survey of the adult population in the UK, conducted in March 2020, revealed that 69.0% of adults would accept a COVID-19 vaccine, 24.8% were uncertain, and 6.1% would refuse [18]. A month later, research across seven European countries showed similar findings with 18.9% of adults being unsure about accepting a COVID-19 vaccine and a further 7.2% not wanting to be vaccinated [19]. A study in the USA, conducted in May 2020, found that 67.0% would accept a COVID-19 vaccine [20]. An Australian study conducted in July 2020, found that 90.0% would accept a COVID-19 vaccination [21]. Identifying the prevalence of vaccine hesitancy, and its associated factors, is necessary to design public educational programs and ensure a successful vaccine roll-out.

We are not aware of any studies on COVID-19 vaccine hesitancy from the Middle East and North Africa (MENA) region that are representative of national populations. Indeed, irrespective of methodology, only three studies on the prevalence of vaccine hesitancy in the MENA region have been published. The first, an online survey (n = 3414) in Arab countries, with 64.0% of the sample from Jordan and 23.0% from Kuwait, showed that only 29.4% of respondents would accept a COVID-19 vaccine when it became available [22]. However, as the study used a convenience sample one cannot generalize the results. The second publication is from the UAE but had a small and non-representative sample (n = 300) [23]. A third study investigated vaccine hesitancy in Qatar, but used a convenience sample (n = 7000) and reported hesitancy of 20.0% [24]. The lack of representative data on vaccine hesitancy from the MENA region is of particular concern as countries within the region have had some of the highest death rates per million population in Africa and Asia. As of April 2021, Libya and Tunisia appeared in the five African countries (n = 57) with the highest death rate per million population with Lebanon and Jordan appearing in the five Asian countries (n = 49) with the highest per capita death rate [25]. Irrespective of geographic differences in infection and mortality rates, combatting COVID-19 requires a global approach.

Given this gap in the literature, we conducted a representative national survey to estimate the prevalence and associated factors of vaccine hesitancy and resistance among the adult population of Qatar.

## 2. Materials and Methods

### 2.1. Setting

With almost 100 nationalities, Qatar is one of the most culturally diverse, richest, and fastest developing countries in the Arabian Peninsula. Most of the 2.9 million population live in the capital, Doha. Although no exact official statistics are available, the majority of the population are young (median age of 30–34) migrant workers (estimated between 80–90%) with the largest subgroup being from the Indian subcontinent [26]. Qatar recorded its first case of COVID-19 in late February 2020 [27]. The following month the government introduced a strict lockdown, restricted entry to the country, and made the wearing of facemasks in public and the use of a smart phone contact-tracing app compulsory. These policies proved effective, the daily rate of infection peaked in late May and fell gradually throughout June and July. Between August 2020 and February 2021, the rate of infection has remained relatively low, with a marked spike in daily infection rates since the beginning of March 2021 potentially marking the “second wave” of the pandemic in this country. Lockdown measures were eased in mid-June 2020, but some have been reinstated at the end of March 2021; social distancing, wearing masks, and using the contact-tracing app remain standard practice. In addition, mandatory quarantine remains a requirement for many travelers entering the country.

Qatar operates a national health service. It has an established national program for childhood vaccinations and seasonal influenza vaccine is offered to high-risk groups. In December 2020, the government announced that it would provide free COVID-19 vaccination to the entire population and that vaccination would occur in stages, starting with the highest priority group. The first phase of vaccination started at the end of December 2020 and continued throughout January and it targeted those aged 70 years and above, people with multiple chronic conditions, and key healthcare staff working in close contact with COVID-19 patients. At the time of writing (April 2021), those being invited for vaccination include people aged 40 years and above, irrespective of their health conditions, people with moderate chronic medical conditions, and key workers in various ministries and industries. Both the Pfizer BioNTech, and Moderna COVID-19 vaccines are approved in Qatar and vaccinations are provided through the country’s Primary Health Care Centers. In addition, two drive-through vaccination centers have been set-up in to administer second doses. In early April 2021, it was announced that Qatar had administered 1 million vaccine doses and that 77 percent of those aged over 60 years had received at least one vaccine dose [28].

### 2.2. Sample Design and Participants

Working with local phone providers, we developed a sampling frame of all cellphones in Qatar and used probability based sampling [29], to select a representative sample of cellphone numbers of Arabic and English speaking adults (18 years of age or older) who confirmed being residents of Qatar and provided verbal informed consent before proceeding with the phone interviews. For more information about the sample, please see Appendix B.

Data collection took place between 15 December 2020 and 25 January 2021 using a phone questionnaire devised by the research team and was made available to participants in either Arabic or English.

### 2.3. Procedures

The Social and Economic Survey Research Institute (SESRI) at Qatar University conducted the phone interviews and data analysis. The survey was conducted by trained researchers on cellular phones using a remote distributed Computer Assisted Telephone Interviewing (CATI) Lab system [30] from 15 December 2020 through 25 January 2021 and entered participants’ responses directly into Blaise survey management software (Blaise, Statistics Netherland). The questionnaire took approximately twenty-five minutes to complete. We applied standardized coding and interpretation procedure for different dialing outcomes and for calculation of response rates [31].

### 2.4. Measures

Sociodemographic characteristics: The questionnaire gathered data on a range of sociodemographic variables including age, gender, marital status, and employment status. Similar to most other countries in the Arabian Gulf, Qatar has three distinct social classes or population groups. The first group is Qatari nationals (QNs) or nationals; most are of tribal origins with shared ancestry and traditions. The second group is higher income white-collar migrants (WCMs) who are highly skilled and educated. The third group is blue-collar migrantss (BCMs) who are mostly young male laborers from South Asia and South East Asia with little or no formal education. Series of questions on citizenship status and income were used to determine classification of respondents into one of the main three population subgroups in Qatar. We considered participants to be BCMs if they reported a combined household income equal to or less than 1100 USD per month while considering those who earned higher than 1100 USD per month as WCMs [32]. Ethnicity or cultural background was determined based on questions in relation to country of origin and language chosen to complete the interview (Arabic versus English), which were then collapsed into Arab versus non-Arab as we were most interested in accounting for cultural differences in attitudes towards the vaccine between main stream culture of Qatar (Arabic) versus other cultures.

Vaccine willingness and vaccine-related questions: With regards to our main dependent variable, participants were asked how strongly they agreed or disagreed with the following statement; “I am willing to get coronavirus vaccine if it became available for me” (5-point Likert Scale: 1 = Strongly agree, 2 = Somewhat agree, 3 = Neutral, 4 = Somewhat disagree, 5 = Strongly disagree). We collapsed responses to this question into three groups: vaccine accepting (strongly agree), vaccine resistant or refusers (strongly disagree) and vaccine hesitant (somewhat agree, neutral, somewhat disagree).

The questionnaire included two other statements, rated on the same 5-point Likert scale, about COVID-19 vaccines: “Getting the coronavirus vaccine should be made mandatory” and “When a coronavirus vaccine becomes available, I will be concerned about its side-effects”.

A further question asked respondents “What would make you more confident in accepting the vaccine?” A range of options were presented and respondents could select only one of the following options: endorsement by ‘my doctor, a public figure, Ministry of Health, World Health Organization (WHO), social media influencer (s), positive feedback from friends or family members, reading scientific research of its effectiveness, and other sources of endorsement’. As a last option in the list, we also provided a statement that read, “I will not accept the vaccine irrespective of endorsement source”.

Pandemic-related questions: The questionnaire included questions about personal history of COVID-19 (reported positive status confirmed by a test), extent of concern about themselves or their family members contracting COVID-19 (not-at-all concerned, not too concerned, somewhat concerned, very concerned), death of someone close due to COVID-19, experience of quarantine since the pandemic started. Additionally, we posed a 5-point Likert scale question about the extent that they agree or disagree with the statement “Measures taken in Qatar have been effective in controlling the spread of the coronavirus’.

Physical health: A positive answer on the following question was taken to indicate the presence of chronic physical disease: “Have you been diagnosed or told by your doctor that you have any of the following conditions?” The list consisted of diabetes, high blood pressure, high cholesterol, asthma, heart disease, and cancer or cancerous tumors, disability (physical, visual, hearing), or “any other long term physical health condition not mentioned?”.

Mental health: We identified participants as having moderate-to-severe symptoms of depression or anxiety in the past two weeks versus no symptoms based on a cut-off of 10 or more on the nine-item Physician Health Questionnaire (PHQ-9) or the 7-items Generalized Anxiety Disorder Questionnaire (GAD-7) [33,34,35,36,37].

## 3. Statistical Analysis

We calculated descriptive statistics including proportions/percentages, mean, standard deviation (SD), and standard errors (SEs) for variables in the study. Chi-square test of proportions were initially used to compare the distribution of socio-demographics, pandemic-related variables, mental- and physical health characteristics across the three categories of vaccine willingness—acceptance, hesitancy, and resistance.

We fitted univariable and multivariable multinomial logistic regression models to identify associations between a number of potential explanatory variables identified from the literature [38,39] and each of vaccine hesitancy and vaccine resistance to receive COVID-19 vaccine relative to acceptance of the vaccine as the reference group. We simultaneously estimated relative risk ratios (RRR) with corresponding 95% confidence intervals (CI) and robust SEs from the exponentiated coefficients of associations between explanatory variables and hesitancy (comparison group) versus acceptance (referent group) and for the same explanatory variables and resistance (comparison group) versus acceptance (referent group). An RRR >1 indicates that the risk of the outcome occurring in the comparison group relative to the risk of the outcome occurring in the referent group increases as the variable increases. An RRR <1 indicates that the risk of the outcome occurring in the comparison group relative to the risk of the outcome occurring in the referent group decreases as the variable increases. An RRR = 1 implies that there is no difference in the risk of the outcome occurring in the comparison group relative to the referent group.

In the univariable models, we entered each potential explanatory variable alone and estimated the unadjusted RRR for each level of the dependent variable relative to the reference group (vaccine acceptance).

We estimated two types of multivariable models. In the fully adjusted model, sociodemographic, pandemic-related, general- and mental- health variables were included. In the reduced model, only variables that were statistically significant at the unadjusted level were included. We examined the contribution of each variable to the final or reduced model using a variety of fit statistics, including the F-adjusted Wald test and the F-adjusted mean residual goodness of fit test.

We conducted statistical analyses using STATA version 16 [40] and statistical significance was defined at an alpha level of 0.05. All our analyses including our models were weighted so that the results are nationally representative.

## 4. Results

Of the 8323 cellphone numbers sampled and were contacted to participate in the study, 1912 cellphone numbers were found eligible including 874 who were eligible but did not participate in the study and 1038 respondents who successfully completed the phone interview (i.e., reaching the last question in the survey), giving an overall response rate adjusted for eligibility of respondents of 44.4% [31]. For more information about sample size, phone interview outcomes, response rate calculation, please see Appendix B.

Table 1 shows the unweighted and weighted sample characteristics. The sample comprised 19.6% Qataris, 57.1% WCMs, and 23.3% BCMs. We also crosschecked these proportions with another survey conducted by SESRI in November of 2019. Unlike our survey, the target population for this survey included everyone who was 18 years or older with no language filtering. We found that among respondents in English or Arabic language, the proportions for QNs, WCMs, and BCMs were 19.8%, 54.4%, and 25.8%, respectively. These proportions are very close (within the sampling errors) to the proportions reported in our survey. Therefore, we are confident that our survey is representative of the English and Arabic speaking population in Qatar. Approximately, one-third of the sample were females and the median age was 38.0 years (SD = 11.2). The majority of respondents were married (68.8%) and employed (75.9%). Approximately, 8.5% met cut-off for moderate-to-severe symptoms of depression or anxiety in the past two weeks from the date of the interview. Approximately, one-quarter (24.5%) had a chronic health condition, 7.9% had suffered from COVID-19, 12.1% reported a person close to them had died of COVID-19, and 22.2% reported have been quarantined since the pandemic started. Participants were “not at all concerned” or “not too concerned” (36.5%), “somewhat concerned” (30.6%), and “very concerned” (32.9%) about themselves or close others contracting COVID-19.

In the total sample, 39.4% strongly agreed and 37.0% somewhat agreed with the statement that the COVID-19 vaccine side effects are of concern and 32.5% strongly agreed that the COVID-19 vaccines should be made mandatory. Most of the sample (81.8%) strongly agreed that effective strategies were taken in Qatar against the spread of COVID-19.

In terms of vaccine willingness, 42.7% (95% CI: 39.5–46.1) were classified as accepting, 45.2% (95% CI: 41.9–48.4) as hesitant, and 12.1% (95% CI: 10.1–14.4) as resistant of COVID-19 vaccination, respectively.

As shown in Figure 1A–C the proportions of participants who strongly agreed that the side-effects of the vaccine are of concern was higher in the vaccine resistant groups (68.8%) compared to the hesitant (40.2%) and accepting groups (30.2%) (*p* < 0.001). Additionally, a larger proportion of participants in the vaccine resistant group (81.5%) compared to participants in the hesitant (22.5%) and accepting (3.7%) groups strongly disagreed with COVID-19 vaccination made mandatory (*p* < 0.001). Furthermore, the proportion who strongly agreed with the statement that measures taken against COVID-19 in Qatar were effective in containing the spread of the virus was higher in the accepting (88.9%) compared to hesitant (76.9%) and resistant groups (75.3%) (*p* < 0.001).

Table 2 shows the result of bivariate analyses. The three vaccine willingness groups differed significantly on a wide range of variables, namely nationality, educational level, gender, age, employment status, language, living arrangements, presence of current anxiety or depression, and COVID-19 related concern. No significant differences were found between the groups in terms of marital status, chronic physical disease, and COVID-19 related variables including experience of quarantine status, COVID-19, or death of a friend or a relative from COVID-19.

As shown in Figure 2, over half of those in the resistant group (58.9%) stated that they would not accept the vaccine regardless of the source of endorsement received compared to 5.6% and 0.2% in the hesitant and accepting groups, respectively (*p* < 0.001). The most important sources of endorsement were the Ministry of Health (MOH) followed by doctors among the hesitant and accepting groups. Smaller proportions of those in the resistant group accepted endorsements by MOH, the WHO, and scientific research compared to the other two groups.

Results from the reduced and the fully adjusted models are similar and are shown in comparison to the unadjusted model in Appendix A.

Table 3 shows the results of the fully adjusted model comparing willingness to receive COVID-19 vaccination with the fifteen variables that we assessed. Only the following variables were significantly associated with increased risk of being vaccine hesitant relative to vaccine accepting, namely female [RRR = 1.57; 95% CI: 1.06–2.33], Arab [RRR = 3.11; 95% CI: 2.15–4.49], BCMs versus Qataris [RRR = 0.49; 95% CI: 0.25–0.94] and those who either somewhat agreed/neutral/somewhat disagreed [RRR = 5.69; 95% CI: 3.06–10.59] or strongly agreed [RRR = 8.28; 95% CI: 4.32–15.90] that the vaccine side-effects are of concern relative to those who strongly disagreed (reference group).

Most of these variables were also significantly associated with increased risk of being vaccine resistant versus accepting including female [RRR = 3.43; 95% CI: 1.87–6.28], Arab [RRR = 4.20; 95% CI: 2.09–8.47], WCMs versus Qataris [RRR = 0.32; 95% CI: 0.16–0.67], BCMs versus Qataris [RRR = 0.32; 95% CI: 0.12–0.87] and being “somewhat” or “very concerned” about themselves or others contracting COVID-19 versus those who were “not too concerned” or “not concerned at all” [RRR = 0.48; 95% CI: 0.27–0.83]. Additionally, only those who strongly agreed versus those who strongly disagreed with the statement that vaccine-side effects are of concern were significantly associated with increased risk of vaccine resistance [RRR = 6.30, 95% CI:2.81–14.13].

## 5. Discussion

This is the first nationally representative study conducted in the MENA with the objectives of estimating prevalence and potential determinants of vaccine hesitancy and resistance, and took place shortly after that the Food and Drug Administration in the USA approved the first COVID-19 vaccine in December of 2020 [9]. The setting is Qatar, one of the richest and most rapidly developing countries in the Arabian Peninsula, a host to over a hundred different nationalities and a country where the population is mostly made up of working-class expatriates from around the world.

Our main findings are that COVID-19 vaccine acceptance is quite low (42.7%) and hesitance (45.2%) is quite high relative to what has been reported to date in other countries including the United Kingdom, Europe, Australia, and the United States [18,19,20,21]. In our study, 12.1% were vaccine refusers i.e., they strongly disagreed with the statement “I am willing to get coronavirus a vaccine if it became available”. This is higher than the 6.1% who said they would refuse a vaccine in the UK [18] and the 7.2% who would refuse a vaccine in a European study [19]. Our vaccine hesitancy rate (45.2%) was higher than 24.8% who answered ‘maybe’ when asked if they would accept a vaccine in a UK study [18] and the 18.9% who were ‘unsure’ about accepting a vaccine in a European study [19]. The corollary is that only 42.7% of our respondents were accepting of vaccination, which is lower than the 69.0% acceptance rate seen in the UK [18].

The fact that less than half of our sample were vaccine accepting is a concern given that a 71–74% immunity level is needed for herd immunity [4,5,6]. A large proportion (76.4%) of our sample strongly or somewhat agreed that the vaccine side effects are of concern—a finding that was high across all three groups: vaccine accepting (68.8%), hesitant (83.8%), and resistant (74.8%) groups. A striking finding in our results is that variables related to actual experience of COVID-19 (i.e., the proportion who had experienced bereavement from COVID-19, had contracted COVID-19, and who had experienced quarantine) did not differ between the three vaccine willingness groups. In contrast, a range of socio-demographic variables distinguished between the three willingness groups and some of these variables emerged as independently associated with vaccine hesitancy and resistance after adjusting for all the other potential explanatory variables.

Our fully adjusted model (Table 3) showed that both hesitancy and resistance, relative to acceptance of the vaccine, were mainly explained by female gender, Arab ethnicity, and concerns about vaccine side-effects. In addition to these variables, Qatari nationality (compared to WCMs and BCMs) was significantly associated with vaccine resistance. Absence of or lack of concern about a family member or themselves contracting COVID-19 compared to those who were “somewhat” or “very concerned” along with strong agreement with concern about vaccine side-effects were the only COVID-19 related variables that were significantly associated with resistance as opposed to acceptance of the vaccine. These findings allow public educational programs to be targeted to those who are more likely to be hesitant or resistant as both of these groups tend to be similar in characteristics.

Our findings share important parallels with work in other parts of the world. Female gender emerged as an independent factor associated with COVID-19 vaccine hesitancy in a study in the USA [20]. Ethnicity or cultural background can influence attitudes towards disease prevention such as vaccination [41]. We found that Arab ethnicity was independently associated with both vaccine resistance and hesitancy. Other studies have assessed ethnicity, though outside the MENA region, and found it to be significantly associated with COVID-19 vaccine hesitancy. For example, in a USA study, acceptance rates were higher in American Indian/Alaska Native, Asian and White groups versus a Black/African American reference group [20]. In Arab culture in particular; fatalistic beliefs have been associated with poor vaccine uptake [42,43,44].

Qatari nationality, versus migrant status, was also associated with increased risk of vaccine resistance. Most of the population of Qatar are working class migrants including BCMs and WCMs whose residency status in the country is largely tied to their employment contracts. Only Qataris have the birthright to stay in the country without employment. The migrant status of WCMs and BCMs means that they are more likely to be accepting of government or employer’s policy, whether it applies to making COVID-19 vaccinations mandatory or other areas.

We found that those who were vaccine resistant differed from those in the vaccine hesitant and vaccine accepting groups in reporting that no source of endorsement will increase their confidence in accepting the COVID-19 vaccine. In contrast, both doctors and the MOH were seen as the two most important sources of information to endorse the vaccine in the hesitant and accepting groups (Figure 2). Both sources of endorsements would need to provide accurate and balanced information and maintain public trust. The vaccine resistant group were twice more likely to endorse their doctor as a source of information than the MOH; in contrast, the vaccine accepting and hesitant groups were slightly more likely to favor the MOH (Figure 2). This implies a greater suspicion of official bodies, in this case, the MOH, by vaccine refusers. Consistent with this, the vaccine refusers in our study were significantly less likely to agree with the statement that the measures taken in Qatar have been effective in controlling the spread of COVID-19 (Figure 1C). Public health officials and doctors, as well as other healthcare professionals, need to be involved in health promotion to encourage vaccine uptake. This requires them to have accurate knowledge of the vaccine and COVID-19. Messages need to be targeted to groups that are more likely to be vaccine resistant including women, Qataris, and Arabs. It is important that messages are culturally sensitive. Accurate information is particularly important as misinformation about vaccines in general [45], and the COVID-19 vaccines [22], is widespread especially on the internet and in social media. Vaccine misinformation can be persuasive [46] and disseminated easily [47], which makes combating it a major challenge. Misinformation about COVID-19 is particularly likely as it is a newly recognized disease, it attracted conspiracy beliefs early in the pandemic including that it was man-made [48] and that vaccines have been developed in record time leading to concerns about safety. In fact, the latter was one of the main reasons cited by those who are COVID-19 vaccine resistant in Australia [21]. To date, evidence shows that the available vaccines are extremely safe. Ensuring a successful vaccine roll-out goes beyond simply providing information. Other key recommendations from an expert group include ensuring vaccination is available in safe, familiar, and convenient places and ensuring that the public has confidence in fair access to vaccines [49].

The main strengths of this study are that it used a probability-based sample from a high coverage frame of the primary telecommunication networks in Qatar. The study also had a relatively high response rate for a phone survey. The data were weighted to account for sampling design and post-stratification to known population targets available from Qatar’s census bureau [50] to reduce residual effects of non-response and under-coverage of the sampling frame. As such the results presented here are representative of the national adult population of Qatar. Though representative, our study may have suffered from under coverage of BCMs due to survey language restriction (Arabic or English). This potentially could limit generalizability of our findings to only English-speaking BCMs. Comparison of vaccine hesitancy across studies can be hindered by the use of different methodologies. Some studies used a single item regarding whether an individual would accept vaccination [18,20], as we did, but other studies have used a scale with several items to assess hesitancy [41]. Even when a single similar item is used, comparisons need to consider different wordings in the stem question and the Likert anchor points. Nevertheless, most studies define a vaccine resistant group as those who clearly state that they would not accept vaccination. Studies vary as to whether or not they subdivide the remainder into those who are vaccine accepting from those who are ambivalent or hesitant about receiving vaccination.

We did not assess the role of religiosity as a determinant of vaccine hesitancy or resistance, which may be an important factor [38,51,52,53,54] especially in conservative countries like Qatar. However, we did assess ethnicity as a proxy measure of culture to which, religiosity is associated. A limitation of our study, and similar studies conducted before COVID vaccines were widely available, is that one cannot assume that the reported view will be mirrored in future behavior. Differences could manifest in either direction, in other words uptake of vaccine may be either lower or greater than earlier studies indicated. Increasing rates of infection and death in several countries, constituting a so-called “second wave”, may convince more people that COVID-19 is a major threat and that vaccination is beneficial. Similarly, positive publicity and a lack of serious side-effects in post marketing surveillance may also help increase vaccination rates. Conversely, the appearance of side-effects and negative publicity could have a detrimental effect.

## 6. Conclusions

Our results indicate a low level of vaccine acceptance (42.7%) and conversely high levels of vaccine hesitancy (45.2%) and vaccine resistance (12.1%) in Qatar. Concerns about side-effects of the vaccine were independently associated with vaccine hesitancy and vaccine resistance. In addition, few socio-demographics namely female gender, Arab ethnicity and non-migrant status were associated with hesitancy and resistance towards COVID-19 vaccine. This suggests that considerable public education may be required to enhance vaccination rates consistent with herd immunity. Our data will assist in targeting educational information to those most likely to be hesitant or resistant. As well as highlighting the efficacy and safety of the vaccine, education needs to highlight the importance of herd immunity i.e., the altruistic benefits of vaccination in protecting vulnerable individuals in the population.

## Figures and Tables

**Figure 1 vaccines-09-00471-f001:**
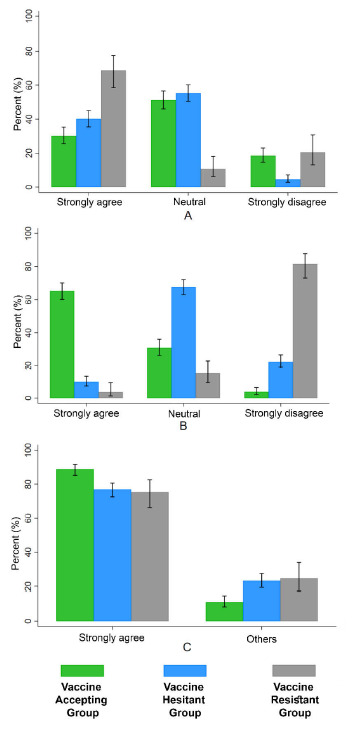
Attitudinal questions related to COVID-19 vaccine and effectiveness of measures taken to contain the virus in Qatar across three vaccine willingness groups: vaccine accepting, vaccine hesitant, and vaccine resistant. (**A**) “When a coronavirus vaccine becomes available, I will be concerned of its side-effects”; (**B**) “Getting the coronavirus vaccine should be made mandatory” (**C**) “Measures taken in Qatar have been effective in controlling the spread of the coronavirus virus”. “Neutral” labels in (**A**,**B**) include somewhat agree, neutral, and somewhat disagree. The label “Others” in (**C**) includes somewhat agree, neutral, and somewhat disagree response categories.

**Figure 2 vaccines-09-00471-f002:**
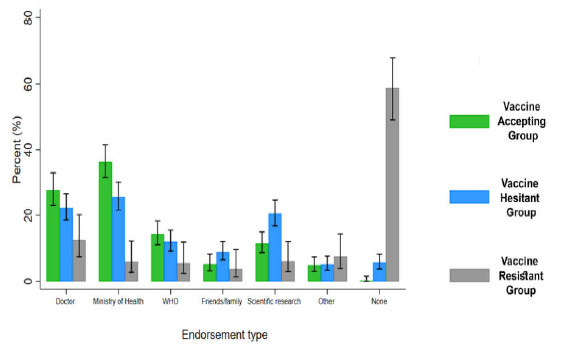
Sources of endorsement to increase confidence in COVID-19 vaccine by three willingness groups: vaccine accepting, vaccine hesitant, and vaccine resistant. Notes. WHO is World Health Organization. “Other” refers to sources of endorsement including social media influencers, public figures, and other sources as elicited by open-ended responses. “None” refers to the response option that “no type of endorsement source will increase in my confidence in accepting the vaccine”.

**Table 1 vaccines-09-00471-t001:** Sample characteristics.

Variables	Frequency (n)	Unweighted Percentages (%)	Weighted Percentages (%)
**Migrant Status/Type**			
Qataris (Non-migrants)	171	16.5	19.6
White-collar migrants	689	66.4	57.1
Blue-collar migrants	178	17.1	23.3
**Education Level**			
Undergraduate or less	895	86.5	87.5
Graduate/Professional	140	13.5	12.5
**Gender**			
Male	709	68.3	66.7
Female	329	31.7	33.3
**Age Group (Years)**			
18–29	201	19.9	23.1
30–34	180	17.8	18.3
35–39	201	19.9	19.1
40+	428	42.4	39.5
**Marital Status**			
Married	759	73.3	68.8
Separated/Divorced/Widowed	48	4.6	5.2
Never married	228	22.0	26.0
**Employment Status**			
Unemployed	240	23.2	24.1
Employed	796	76.8	75.9
**Ethnicity**			
Arab	579	55.8	54.1
Non-Arab	459	44.2	45.9
**Living Arrangement**			
Live with Others	863	83.1	83.0
Live Alone	175	16.9	17.0
**Depression or Anxiety ^1^**			
Yes	81	8.02	8.5
No	928	92.0	91.5
**Chronic Disease ^2^**			
Yes	268	25.8	24.5
No	770	74.2	75.5
**COVID-19 Status ^3^**			
Yes	85	8.2	7.9
No	953	91.8	92.1
**COVID-19 Related Death ^4^**			
Yes	134	12.9	12.1
No	904	87.1	87.9
**Quarantine Status ^5^**			
Yes	229	22.1	22.2
No	809	77.9	77.8
**COVID-19 Infection Concerns ^6^**			
Not at all concerned	184	19.5	20.4
Not too concerned	159	16.8	16.1
Somewhat concerned	307	32.5	30.6
Very concerned	295	31.2	32.9
**Willing to Get the Vaccine ^7^**			
Accepting Group	430	42.0	42.7
Hesitant Group	475	46.4	45.2
Resistant Group	118	11.5	12.1

Note. Total sample size *N* = 1038. Weighted percentages are calculated using survey weights and therefore differ from the unweighted or raw percentages. The number of respondents (n) reported for each variable corresponds to the unweighted sample. ^1^ Defined as a cut-off of 10 or more on Physician Health Questionnaire (PHQ-9) or the Generalized Anxiety Disorder Questionnaire (GAD-7); ^2^ Chronic disease defined by respondents’ endorsement of one of the following conditions: diabetes, high blood pressure, high cholesterol, asthma, heart disease, cancer or cancerous tumors, and disability (physical, visual, hearing); ^3^ Defined as positive if the respondent reported positive status confirmed by a test; ^4^ Defined as the death of someone close due to COVID-19; ^5^ Defined as any experience of quarantine since the pandemic started; ^6^ Measuring the extent of concern about themselves or their family members contracting COVID-19 (not at all concerned, not too concerned, somewhat concerned, very concerned); ^7^ Willingness to get the vaccine was categorized into: vaccine accepting (strongly agree), vaccine hesitant (somewhat agree, neutral, somewhat disagree), and vaccine resistant (strongly disagree).

**Table 2 vaccines-09-00471-t002:** Willingness to get COVID-19 vaccine by socio-demographics, work-related, and health-related characteristics.

Explanatory Variables		Willingness to Get the Vaccine ^11^ (%)			
		Accepting	Hesitant	Resistant	*p*-Value *
**Migrant Status/Type**	Qataris (non-migrants)	10.8	21.1	43.5	<0.0001
	White-collar migrants	57.8	61.7	41.8	
	Blue-collar migrants	31.4	17.2	14.7	
**Education Level**	Undergraduate or less	84.5	88.0	94.7	0.010
	Graduate/Professional	15.5	12.0	5.3	
**Gender**	Male	77.5	61.5	44.9	<0.0001
	Female	22.5	38.5	55.1	
**Age Group (Years)**	18–29	20.9	22.6	33.4	0.003
	30–34	15.8	23.2	8.8	
	35–39	19.7	19.1	16.9	
	40+	43.7	35.6	40.9	
**Marital Status**	Ever married	73.3	74.9	74.8	0.889
	Never married	26.7	25.1	25.2	
**Employment Status**	Unemployed	15.8	28.0	38.4	<0.0001
	Employed	84.2	72.0	61.6	
**Ethnicity**	Arab	36.6	65.0	76.0	<0.0001
	Non-Arab	63.4	35.0	24.0	
**Living Arrangement**	Live with others	77.1	87.1	87.6	0.001
	Live alone	22.9	12.9	12.4	
**Depression or Anxiety ^1^**	Yes	5.8	9.6	14.3	0.018
	No	94.2	90.4	85.7	
**Chronic Disease ^2^**	Yes	24.6	22.7	32.1	0.125
	No	75.4	77.3	67.9	
**COVID-19 Status ^3^**	Yes	7.8	7.8	9.5	0.819
	No	92.2	92.2	90.5	
**COVID-19 Related Death ^4^**	Yes	11.9	13.7	8.9	0.337
	No	88.1	86.3	91.1	
**Quarantine Status ^5^**	Yes	23.0	22.3	20.5	0.848
	No	77.0	77.7	79.5	
**COVID-19 Infection Concerns ^6^**	Not at all/Not too concerned	33.1	35.3	50.0	<0.0001
	Somewhat concerned	25.6	37.3	25.8	
	Very concerned	41.3	27.4	24.2	
**Effective COVID-19 Containment in Qatar ^7^**	Strongly agree	88.9	76.9	75.3	<0.0001
	Somewhat agree or disagree/Neutral	11.1	23.2	24.7	
	Strongly agree	30.2	40.2	68.8	<0.0001
**COVID-19 Vaccine Side-Effects Are of Concern ^8^**	Somewhat agree	38.6	43.6	6.0	
	Neutral	1.9	2.2	1.0	
	Somewhat disagree	10.8	9.5	3.8	
	Strongly disagree	18.5	4.5	20.5	
**COVID-19 Vaccine Should Be Mandatory ^9^**	Strongly agree	65.3	9.8	3.6	<0.0001
	Somewhat agree	21.3	34.6	5.0	
	Neutral	1.7	5.0	1.5	
	Somewhat disagree	8.0	28.1	8.4	
	Strongly disagree	3.7	22.5	81.5	
**Endorsement Source for Vaccine ^10^**	My doctor	27.7	22.3	12.5	<0.0001
	Ministry of Health	36.4	25.7	5.9	
	WHO	14.3	12.0	5.4	
	Positive feedback from Friends/Family	5.2	8.9	3.7	
	Scientific research	11.5	20.5	6.0	
	Other	4.8	5.1	7.6	
	I will not accept vaccine	0.2	5.6	58.9	

Notes. All reported percentages % are weighted. Columns add up to 100%. * Probabilities (*p*-values) were derived from design-based *F* test statistics, a weighted Pearson chi square statistic corrected for complex sampling design. The null hypothesis tests the assumption that there is no association between each explanatory variable with COVID-19 vaccine willingness groups. A test of *p*-value less than 0.05 would mean that the null hypothesis of independence between the explanatory variable and COVID-19 vaccine willingness groups could be rejected. The corresponding inference, in this case, would support evidence of statistical association between the explanatory variable and COVID-19 vaccine willingness groups. ^1^ Defined as a cut-off of 10 or more on Physician Health Questionnaire (PHQ-9) or the Generalized Anxiety Disorder Questionnaire (GAD-7); ^2^ Chronic disease defined by respondents’ endorsement of one of the following conditions: diabetes, high blood pressure, high cholesterol, asthma, heart disease, cancer or cancerous tumors, and disability (physical, visual, hearing); ^3^ Defined as positive if the respondent reported positive status confirmed by a test; ^4^ Defined as death of someone close due to COVID-19; ^5^ Defined as any experience of quarantine since the pandemic stared; ^6^ Measuring the extent of concern about themselves or their family members contracting COVID-19 (not at all concerned, not too concerned, somewhat concerned, very concerned); ^7^ Defined as the extent of agreement/disagreement that the measures taken in Qatar have been effective in controlling the spread of the coronavirus; ^8^ Defined as the degree of agreement/disagreement that “When a coronavirus vaccine becomes available, I will be concerned of its side effects”; ^9^ Defined as the degree of agreement/disagreement that “getting the coronavirus vaccine should be made mandatory”; ^10^ Based on the question “What would make you more confident in accepting the vaccine?”; ^11^ Defined as the degree of agreement/disagreement with the statement “I am willing to get coronavirus vaccine if it became available for me”. See text for details of how 5-point Likert Scale was collapsed into accepting, hesitant, and resistant groups.

**Table 3 vaccines-09-00471-t003:** Model of the associations between willingness to get COVID-19 vaccine and socio-demographics, work—and health-related characteristics.

Variables	Reference Category	Fully Adjusted Model Willingness to Get Vaccine					
		Hesitant Versus Accepting			Resistant Versus Accepting		
		RRR	95% CI	*p*-Value	RRR	95% CI	*p*-Value *
**Age Group**	18–29						
30–34		1.72	0.98–3.03	0.061	0.69	0.26–1.80	0.443
35–39		1.16	0.65–2.07	0.622	0.83	0.33–2.08	0.696
40+		0.72	0.42–1.27	0.263	0.59	0.25–1.41	0.236
**Gender**	Male						
Female		1.57	1.06–2.33	0.023	3.43	1.87–6.28	<0.0001
**Migrant Status/Type** White-collar migrants	Qataris (non-migrant)	0.68	0.39–1.17	0.163	0.32	0.16–0.67	0.002
Blue-collar migrants		0.49	0.25–0.94	0.032	0.32	0.12–0.87	0.025
**Ethnicity** Arab	Non-Arab	3.11	2.15–4.49	<0.0001	4.20	2.09–8.47	<0.0001
**Education level**	Undergrad or less						
Graduate/Professional		0.78	0.50–1.21	0.270	0.44	0.16–1.21	0.111
**Employment status**	Unemployed						
Employed		0.94	0.60–1.47	0.779	1.18	0.61–2.31	0.619
**Marital status**	Ever married						
Never married		0.89	0.55–1.41	0.614	1.07	0.51–2.26	0.850
**Living arrangement**	Live with Others						
Live alone		0.68	0.44–1.05	0.083	1.08	0.50–2.35	0.843
**Depression or anxiety ^1^** Yes	No	1.10	0.57–2.11	0.778	1.30	0.54–3.09	0.558
**Chronic disease ^2^** Yes	No	0.83	0.56–1.29	0.351	1.16	0.62–2.16	0.637
**Quarantine status ^3^** Yes	No	0.82	0.54–1.23	0.340	0.54	0.26–1.11	0.096
**COVID-19 related death ^4^** Yes	No	1.18	0.73–1.88	0.500	0.66	0.26–1.66	0.380
**COVID-19 infection concerns ^5^**							
Somewhat/Very concerned	Not concerned at all/Not too concerned	1.01	0.72–1.43	0.938	0.48	0.27–0.83	0.009
**COVID-19 vaccine side-effects are of concern ^6^**							
Somewhat agree or disagree/Neutral Strongly agree	Strongly disagree	5.69 8.28	3.06–10.59 4.32–15.90	<0.0001 <0.0001	0.53 6.30	0.21–1.34 2.81–14.13	0.178 <0.0001

Notes. CI is the 95% confidence interval. * Probabilities (*p*-values) were derived from design-based Wald *F* test statistics, a weighted Pearson chi square statistic corrected for complex sampling design. RRR is relative risk ratio. Fully adjusted model of the association between willingness to take the vaccine and all variables (*n* = 849) with the exception of COVID-19 status as it was collinear with concerns about oneself/others contracting COVID-19. ^1^ Defined as a cut-off of 10 or more on Physician Health Questionnaire (PHQ-9) or the Generalized Anxiety Disorder Questionnaire (GAD-7); ^2^ Chronic disease defined by respondents’ endorsement of one of the following conditions: diabetes, high blood pressure, high cholesterol, asthma, heart disease, cancer or cancerous tumors, and disability (physical, visual, hearing); ^3^ Defined as any experience of quarantine since the pandemic started; ^4^ Defined as the death of someone close due to COVID-19; ^5^ Measuring the extent of concern about themselves or their family members contracting COVID-19 (not at all concerned, not too concerned, somewhat concerned, very concerned); ^6^ Defined as the degree of agreement/disagreement that “When a coronavirus vaccine becomes available, I will be concerned of its side effects”.

## Data Availability

Data will be shared upon written request from corresponding author.

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
