# Peer review of "Prevalence and Potential Determinants of COVID-19 Vaccine Hesitancy and Resistance in Qatar: Results from a Nationally Representative Survey of Qatari Nationals and Migrants between December 2020 and January 2021"

_vaccines, 2021, doi:10.3390/vaccines9050471_

Round 1

Reviewer 1 Report

Estimated Authors,

Estimated Editors,

I'm congratulating with Khaled et al. for the very high quality of this paper on the Prevalence and Potential Determinants of COVID-19 Vaccine Hesitancy and Resistance in Qatar. Authors have identified a somewhat worrisom prevalence of vaccine opposition/hesitancy among Qatari residents (noteworthly, assessed through a population-representative sample), and particularly in certain population groups.

Methods and Results are properly reported, and no adjustements are required.

I've only a couple of recommendations for improving the yet very high quality of this paper.

1) table 2-3: please report alogside "p-value" the correspondent test(s) you performed in order to obtain that value;

2) discussion should be a little bit reframed. Please begin with a short summary of your results rather than with the limits of your study, that should be moved at the end.

Author Response

I'm congratulating with Khaled et al. for the very high quality of this paper on the Prevalence and Potential Determinants of COVID-19 Vaccine Hesitancy and Resistance in Qatar. Authors have identified a somewhat worrisom prevalence of vaccine opposition/hesitancy among Qatari residents (noteworthly, assessed through a population-representative sample), and particularly in certain population groups.

Authors’ response: Thank you.

Methods and Results are properly reported, and no adjustements are required.

Authors’ response: Thank you.

I've only a couple of recommendations for improving the yet very high quality of this paper.

1) table 2-3: please report alogside "p-value" the correspondent test(s) you performed in order to obtain that value;

Authors’ response: As per your suggestion, we have added this information in the revised document as notes at the bottom of Table 2 (lines 333-334) and Table 3 (lines 383-384).

2) discussion should be a little bit reframed. Please begin with a short summary of your results rather than with the limits of your study, that should be moved at the end.

Authors’ response: We have implemented these suggestions. Please see first paragraph (lines 404-410) and last three paragraphs (lines 510-515, 517-528, and 529-540) in the discussion of the revised manuscript.

Reviewer 2 Report

The authors surveyed the attitude of various populations in Qatar towards COVID-19 vaccine and investigated the demographic characteristics, social and economic status, health factors and external endorsements that may influence the vaccine acceptance.   Information provided in the paper will be important and useful not only for Qatar but also for countries with similar culture and demographic makeup in designing vaccine and vaccine promoting campaigns.  Authors studied effectiveness of endorsement from health care profession, governmental health ministry, family and other organizations in influencing the vaccine acceptance, however, they should also include religious organization in the evaluation. 

There are some issues need to be addressed. 

  1. Lines 231-232, it is not clear why in 8500 contacted only 4572 were eligible. Based on the criteria in sample design and participants section, it appears that all people who were residents and 18 years of age or older would be eligible.   Authors please explain what the exclusion criteria was.  Is language the only exclusion criteria? 
  2. Authors indicated that 1038 of 4572 eligible persons completed the survey. Assuming many of the eligible persons only partially completed the survey, did authors examine the partial survey results to see if the complete survey population and incomplete survey populations were significantly different in demographic characteristics and attitude to COVID-19 vaccine?  Please discuss. 
  3. Table 1, proportion of WCM participants appeared significantly higher than either QNs or BCMs. Is it reflecting the proportion populations in Qatar? Or is there any possible sampling bias?  Please discuss. 
  4. Table 2, the authors presented proportions of different demographic characteristics, social economic status and COVID related characteristics in three COVID vaccine acceptance groups. It seems that the more direct comparison should be the proportions of three different COVID vaccine acceptance groups in each of the characteristic categories.  For instance, the proportions of accept, hesitant and resistant groups in Qataris (possibly 14%, 28% and 58%) should be calculated and compared to the expected proportion of 33.3% if attitudes towards vaccine were randomly distributed among Qataris.  Or these proportions can be compared to other demographic groups.  Authors should either modify the table 2 or explain why their presentation would be more relevant to the purpose of the study. 
  5. To benefit the average readers, the authors should indicate these p-values in table 2 represented what comparison; what was the null hypothesis and what was the conclusion for the statistical significant result.
  6. Figure 2, the authors did not include the endorsement of religious organization as possible influence of vaccine acceptance. Given the high resistant rate in Qataris population religion might be an important factor as authors suggested.  Authors please discuss why religious factors were not included in the analysis and if omission of religious factor may affect the results of the study.  
  7. Table 3, authors should add explanation what RRR > 1 or < 1 mean for average readers and how high or low of RRR is biologically relevant.   
  8. In discussion section, authors should refer the RRR numbers when discuss results related to their multivariable analysis results in table 3 so that average readers will have better understanding of the analyses.

Minor points

  1. Authors should indicate the statistical significant differences in their figures.
  2. Line 188, ‘not to concerned’ should be ‘not too concerned’

Author Response

Comments and Suggestions for Authors

The authors surveyed the attitude of various populations in Qatar towards COVID-19 vaccine and investigated the demographic characteristics, social and economic status, health factors and external endorsements that may influence the vaccine acceptance.   Information provided in the paper will be important and useful not only for Qatar but also for countries with similar culture and demographic makeup in designing vaccine and vaccine promoting campaigns.  Authors studied effectiveness of endorsement from health care profession, governmental health ministry, family and other organizations in influencing the vaccine acceptance, however, they should also include religious organization in the evaluation. 

There are some issues need to be addressed. 

  1. Lines 231-232, it is not clear why in 8500 contacted only 4572 were eligible. Based on the criteria in sample design and participants section, it appears that all people who were residents and 18 years of age or older would be eligible.   Authors please explain what the exclusion criteria was.  Is language the only exclusion criteria?

Authors’ response: We correctly reported the information about the sample size (8,323), the number of eligible (1,912) and completed interviews (1,038) in the appendix. However, in the first paragraph of section 4, we did not report some of the information correctly. In the revised version of the paper, we have fixed this reporting error. Please see lines 261-264 in the revised manuscript.

As you correctly noticed, the number of eligible phone numbers is much smaller than the total number of phone numbers contacted. This is so because many phone numbers are non-working numbers (i.e. disconnected phone line), people less than 18 years old, or business phone numbers. In addition, in Qatar and in other GCC countries, there is a large population of low-income migrants from the Indian subcontinent (including India, Bangladesh, Nepal, Pakistan) who cannot speak English or Arabic. These people are not eligible for our survey. This explains the big gap between the number of eligible and the total sample size.

With respect to the inclusion criteria, this is clearly stated in section 2.2: 18 years or older, resident of Qatar and speaks Arabic or English. As such people who did not meet any one of these criteria were excluded (Lines 154-162).

  1. Authors indicated that 1038 of 4572 eligible persons completed the survey. Assuming many of the eligible persons only partially completed the survey; did authors examine the partial survey results to see if the complete survey population and incomplete survey populations were significantly different in demographic characteristics and attitude to COVID-19 vaccine?  Please discuss.

Authors’ response: As mentioned above, we corrected the number of eligible participants. It is now 1,912, almost twice the number of completed interviews (1,038). Many of these interviews were “break-offs” (occur when a respondent stops the interview in-progress) that did not progress far enough in the interview to reach sociodemographic section. Therefore, these cases were mostly missing information about their demographic characteristics and attitude questions about the vaccine. To account for possible bias from this non-response, we calculated the propensity weights and applied them in our analysis. This (use of propensity weights) is the standard technique used in the survey industry to address the non-response bias.

  1. Table 1, proportion of WCM participants appeared significantly higher than either QNs or BCMs. Is it reflecting the proportion populations in Qatar? Or is there any possible sampling bias?  Please discuss. 

Authors’ response: There is no publication from Qatar Statistical Authority (PSA) about the proportion of QNs, WCMs, and BCMs among English and Arabic speaking population. However, we know that QNs should be the smallest group. According to the last Government Labor Force Survey conducted in September 2020, QNs only accounts for 9% of the population 18 years or older. Of course, this number will be higher after we exclude a large number of people from Indian subcontinent who cannot speak English or Arabic (as mentioned above).

We also crosschecked the proportions in our survey with another survey conducted by SESRI in November of 2019. Unlike our survey, the target population for that survey included everyone who was 18 years or older with no language filtering. We found that among respondents in English or Arabic language, the proportions for QNs, WCMs, and BCMs are 19.8%, 54.4%, and 25.8%, respectively. These proportions are very close (within the sampling errors) to the proportions reported in our survey. Therefore, we are confident that our survey are representative of the English and Arabic speaking population in Qatar. We also added a footnote with this explanation to clarify this to the reader, please see footnote on page 6 of the revised manuscript.

  1. Table 2, the authors presented proportions of different demographic characteristics, social economic status and COVID related characteristics in three COVID vaccine acceptance groups. It seems that the more direct comparison should be the proportions of three different COVID vaccine acceptance groups in each of the characteristic categories.  For instance, the proportions of accept, hesitant and resistant groups in Qataris (possibly 14%, 28% and 58%) should be calculated and compared to the expected proportion of 33.3% if attitudes towards vaccine were randomly distributed among Qataris.  Or these proportions can be compared to other demographic groups.  Authors should either modify the table 2 or explain why their presentation would be more relevant to the purpose of the study.

Authors’ response:  In Table 2, we chose to present and compare the distribution of different sociodemographics across three main groups as this approach is more aligned with the main objectives of the study, which is to estimate prevalence of the three different COVID acceptance groups and identify factors associated with these vaccine willingness groups. For example, when we report as we did that approximately 42.7% (95%CI: 39.5-46.1) of our sample were classified as accepting, then it would make more intuitive sense to also report what percentage of this accepting group are Qataris (10.8%), are White-collar (57.8%) and Blue-collar (31.5%) migrants as per Table 2. Additionally, this approach would also make comparisons across these three main vaccine groups more intuitive.  As per current presentation in Table 2, for example, we could easily see that the distribution of Qataris is lowest in the accepting group (10.8%), followed by hesitant group (21.1%), but largest in the resistant group, where the make up approximately half of that sample (43.5%).  Having now explained the rationale behind our approach, we also prepared a version of Table 2 (Supplementary Table2) where the presentation of the proportions of three different COVID vaccine acceptance groups is in each of the sociodemographic categories as per your suggestion in case some readers find this presentation more intuitive.  

  1. To benefit the average readers, the authors should indicate these p-values in table 2 represented what comparison; what was the null hypothesis and what was the conclusion for the statistical significant result.

Authors’ response: Thank you for the suggestion. We have now added this information to the notes at the bottom of table 2. Please see lines 334 to 338 in the revised manuscript.

  1. Figure 2, the authors did not include the endorsement of religious organization as possible influence of vaccine acceptance. Given the high resistant rate in Qataris population religion might be an important factor as authors suggested.  Authors please discuss why religious factors were not included in the analysis and if omission of religious factor may affect the results of the study.  

Authors’ response: Thank you for your comment. We agree that religion may well be an important factor that could influence vaccine acceptance, hesitancy or resistance in our context. However, participants under “other” response option for this question did not report it. Although, religion was not directly measured in our study, we indirectly accounted for some of its effects by controlling for ethnicity/cultural background. This is by no means an ideal approach and now we acknowledge this as limitation of our study. Please see lines 529-532 in the revised manuscript.

  1. Table 3, authors should add explanation what RRR > 1 or < 1 mean for average readers and how high or low of RRR is biologically relevant.   

Authors’ response: Thank you for your suggestion, these explanations of the meaning of RRR have been added to the statistical analysis section, please see lines 240-246 in the revised manuscript.

  1. In discussion section, authors should refer the RRR numbers when discuss results related to their multivariable analysis results in table 3 so that average readers will have better understanding of the analyses.

Authors’ response: All the numbers in Table 3 have already been reported in the results section and would therefore be redundant to repeat them again in the discussion. We also refer the reader to Table 3 in the discussion to guide them to the source of this information, while avoiding repetition.

Minor points

  1. Authors should indicate the statistical significant differences in their figures.

Authors’ response: Thank you for the suggestion. These differences in significance were already reported in the results section following each figure:  lines 306-314 for Figure 1 and lines 350-356 for Figure 2. Therefore, it would be redundant to include them in the figures as well. Alternatively, we included the 95% confidence intervals in the figures (which were not reported in the text) as these are more visually informative to readers in terms of both magnitude and precision of the estimates as well as inference about significance (overlapping versus non-overlapping intervals).

  1. Line 188, ‘not to concerned’ should be ‘not too concerned’

Authors’ response: Thank you, this typo has been corrected. See line 211 in the revised manuscript

Reviewer 3 Report

The study was performed to measure the prevalence of hesitancy and resistance of the COVID-19 vaccine in Qatar. Indeed, it is a critical study to fill the gap of what determines and the hesitancy of the COVID19 vaccine and the prevalence of the same. The data was recorded by phone survey asking a set of questions. Although authors cover a broad spectrum of questions, I wonder if the questions are validated. Also, the response rate of 44.4% is relatively low than most phone-based surveys. Overall, the data appear to support the conclusion of low acceptance and high hesitancy and resistance to the COVID19 vaccine.

Specific Comments

  1. Consider mentioning the specific data points, e.g., line 92: “most respondents” mention the percent of respondents from Jordon and Kuwait. Similar changes in line 94: “lowest acceptance rate,” line 99: “highest death rates per million.”
  2. Line: 210,211, and 212 Consider giving reference for “variables from literature” line: 210,211 and 212
  3. Consider reformatting the data figures as they seem to be stretched and difficult to read.
  4. Out of curiosity, why use RRR instead of odds ratio in Table 3.
  5. Consider mentioning the p-Value in the figure or figure legend. Also, it would be nice to have explanatory legends.

Author Response

Comments and Suggestions for Authors

The study was performed to measure the prevalence of hesitancy and resistance of the COVID-19 vaccine in Qatar. Indeed, it is a critical study to fill the gap of what determines and the hesitancy of the COVID19 vaccine and the prevalence of the same. The data was recorded by phone survey asking a set of questions. Although authors cover a broad spectrum of questions, I wonder if the questions are validated. Also, the response rate of 44.4% is relatively low than most phone-based surveys. Overall, the data appear to support the conclusion of low acceptance and high hesitancy and resistance to the COVID19 vaccine.

Specific Comments

  1. Consider mentioning the specific data points, e.g., line 92: “most respondents” mention the percent of respondents from Jordon and Kuwait. Similar changes in line 94: “lowest acceptance rate,” line 99: “highest death rates per million.”

Authors’ response: We have revised the text to address your concerns. Please see the 4th paragraph in the revised introduction lines 100-101 and 110-114.

  1. Line: 210,211, and 212 Consider giving reference for “variables from literature” line: 210,211 and 212

Authors’ response: Thank you for your suggestion. We have added two references to support this statement. Please see references 38 and 39 (line 235) of the revised manuscript.

  1. Consider reformatting the data figures as they seem to be stretched and difficult to read.

Authors’ response: Thank for your suggestion. We have reformatted the figures - increased size and resolution. For Figure 1, we also changed the orientation. 

  1. Out of curiosity, why use RRR instead of odds ratio in Table 3.

Authors’ response: RRR or relative risk ratio rather than OR or odds ratio is the default statistics for fitting multinomial models in the statistical package that we used for our analysis (STATA). The RRR is the exponentiated value of a coefficient from the multinomial logistic model – a model where the dependent variable has more than two categories and these categories are considered nominal – i.e. of no ordinal value.  This unordered categorical property of dependent variable distinguishes the multinomial logistic regression from the binary logistic regression (which is appropriate for outcome with two categories, which can be thought of as ordered). By this definition, the exponentiated value of a coefficient from such model represents more closely the relative-risk ratio for a one-unit change in the corresponding variable where risk is measured as the risk of the outcome relative to the base outcome rather than the ratio of two odds or odds ratio obtained from logistic regression. Presenting RRR or OR stems more from disciplinary preference - most disciplines do not distinguish between the two types of measures of association.

  1. Consider mentioning the p-Value in the figure or figure legend. Also, it would be nice to have explanatory legends

Authors’ response:  the p-values were already reported in the text of the results section following each figure:  lines 306-314 for Figure 1 and lines 350-356 for Figure 2. Therefore, it would be redundant to include them in the figures as well. Alternatively, we included the 95% confidence intervals in the figures (which were not reported in the text) as these are more visually informative to readers in terms of both magnitude and precision of the estimates as well as inference about significance (overlapping versus non-overlapping intervals). Concerning the figure legends, we included more explanatory notes at the bottom of Figure 2 in the revised manuscript, which were accidently omitted from the original submission